# Role of iRhom2 in Olfaction: Implications for Odorant Receptor Regulation and Activity-Dependent Adaptation

**DOI:** 10.3390/ijms25116079

**Published:** 2024-05-31

**Authors:** Stephanie A. Azzopardi, Hsiu-Yi Lu, Sebastien Monette, Ariana I. Rabinowitsch, Jane E. Salmon, Hiroaki Matsunami, Carl P. Blobel

**Affiliations:** 1Weill Cornell Medicine/Rockefeller University/Memorial Sloan-Kettering Cancer Center, Tri-Institutional MD-PhD Program, New York, NY 10021, USA; sta2017@med.cornell.edu (S.A.A.); arr2013@med.cornell.edu (A.I.R.); 2Physiology, Biophysics and Systems Biology Program, Weill Cornell Medicine, New York, NY 10021, USA; 3Arthritis and Tissue Degeneration Program, Hospital for Special Surgery, New York, NY 10021, USA; 4Department of Molecular Genetics and Microbiology, Duke University School of Medicine, Durham, NC 27710, USA; justice.lu@duke.edu; 5Tri-Institutional Laboratory of Comparative Pathology, Hospital for Special Surgery, Memorial Sloan Kettering Cancer Center, The Rockefeller University, Weill Cornell Medicine, New York, NY 10021, USA; monettes@mskcc.org; 6Department of Biochemistry, Cellular and Molecular Biology, Weill Cornell Medicine, New York, NY 10021, USA; 7Autoimmunity and Inflammation Program, Hospital for Special Surgery, New York, NY 10021, USA; 8Department of Medicine, Weill Cornell Medicine, New York, NY 10021, USA; 9Department of Neurobiology, Duke Institute for Brain Sciences, Duke University, Durham, NC 27710, USA

**Keywords:** iRhom2 (inactive Rhomboid-like protein 2), ADAM17 (a disintegrin and metalloprotease 17), G-protein coupled receptor (GPCR), olfactory sensory neuron (OSN), olfactory receptor (OR), olfactory epithelium (OE), RNAseq, RNAScope in situ hybridization (ISH), single-cell RNAseq analysis

## Abstract

The cell surface metalloprotease ADAM17 (a disintegrin and metalloprotease 17) and its binding partners iRhom2 and iRhom1 (inactive Rhomboid-like proteins 1 and 2) modulate cell–cell interactions by mediating the release of membrane proteins such as TNFα (Tumor necrosis factor α) and EGFR (Epidermal growth factor receptor) ligands from the cell surface. Most cell types express both iRhoms, though myeloid cells exclusively express iRhom2, and iRhom1 is the main iRhom in the mouse brain. Here, we report that iRhom2 is uniquely expressed in olfactory sensory neurons (OSNs), highly specialized cells expressing one olfactory receptor (OR) from a repertoire of more than a thousand OR genes in mice. *iRhom2-/-* mice had no evident morphological defects in the olfactory epithelium (OE), yet RNAseq analysis revealed differential expression of a small subset of ORs. Notably, while the majority of ORs remain unaffected in *iRhom2-/-* OE, OSNs expressing ORs that are enriched in *iRhom2-/-* OE showed fewer gene expression changes upon odor environmental changes than the majority of OSNs. Moreover, we discovered an inverse correlation between the expression of iRhom2 compared to OSN activity genes and that odor exposure negatively regulates iRhom2 expression. Given that ORs are specialized G-protein coupled receptors (GPCRs) and many GPCRs activate iRhom2/ADAM17, we investigated if ORs could activate iRhom2/ADAM17. Activation of an olfactory receptor that is ectopically expressed in keratinocytes (OR2AT4) by its agonist Sandalore leads to ERK1/2 phosphorylation, likely via an iRhom2/ADAM17-dependent pathway. Taken together, these findings point to a mechanism by which odor stimulation of OSNs activates iRhom2/ADAM17 catalytic activity, resulting in downstream transcriptional changes to the OR repertoire and activity genes, and driving a negative feedback loop to downregulate iRhom2 expression.

## 1. Introduction

Inactive Rhomboid 2 (iRhom2) and the related iRhom1 are seven membrane-spanning proteins that regulate the function of the cell surface metalloprotease a disintegrin and metalloprotease 17 (ADAM17) [1,2,3,4,5,6,7,8,9,10,11]. Both iRhoms are widely expressed, with two notable exceptions: iRhom1 expression is low or absent in myeloid cells, and iRhom2 expression is low or absent in the brain in mice, except for microglia [5,12]. ADAM17 has important roles in development and disease by regulating the TNFα-, EGFR- and other signaling pathways [12,13,14,15,16,17]. Since iRhom2 is required for the activity of ADAM17 in myeloid cells, mice lacking iRhom2 are protected from endotoxin shock and inflammatory arthritis [1,12], like mice lacking ADAM17 in myeloid cells [12]. Moreover, a double knockout mouse strain lacking both iRhom1 and 2 resembles ADAM17-knockout mice in that they die shortly after birth and have open eyes, heart valve defects and growth plate defects [5]. Proteomic analysis has discovered substrates for iRhom1 in the mouse brain [18], yet little is currently known about the role of iRhom2 in the nervous system.

Using an iRhom2-LacZ reporter mouse, we identified prominent iRhom2 expression in olfactory sensory neurons (OSNs), which was unexpected due to the otherwise low iRhom2 expression in the mouse brain. OSNs, unique neurons undergoing lifelong regeneration with a typical lifespan of six to eight weeks, detect odors via activation of olfactory receptors (ORs) and are directly exposed to the external environment, unlike the rest of the brain [19]. ORs, the largest family of G-protein coupled receptors (GPCRs), comprise about 400 genes in humans and about 1200 in mice [20]. During development, immature OSNs co-express multiple ORs, then undergo positive and negative selection, resulting in mature OSNs expressing a single OR type [21,22]. Furthermore, the OR composition in the olfactory epithelium (OE) undergoes activity-dependent sculpting, allowing neurogenesis and/or OSN survival to be modulated based on activity [23,24]. This process, often studied through unilateral naris occlusion in mice, enables adaptation to specific olfactory environments by adjusting the population of OSNs expressing individual ORs.

The selective iRhom2 expression in OSNs prompted questions about its OE function. We conducted histopathological analyses of the OE of *iRhom2-/-* mice to assess changes in OSN maturation or turnover. We used an unbiased RNAseq approach to examine how iRhom2 inactivation affects OR and non-OR expression in the OE, focusing on activity-dependent genes. Additionally, we probed whether OR signaling activates iRhom2/ADAM17 by utilizing the HaCaT keratinocyte cells line, which endogenously expresses the receptor OR2AT4. Our findings provide evidence for a role of iRhom2 in OR landscape regulation and activity-dependent OSN expression programs.

## 2. Results

### 2.1. iRhom2 Is Prominently Expressed in the Olfactory Sensory Neurons

Previous studies have demonstrated that iRhom2 is required for the maturation of ADAM17 and TNFα release from microglia but that it cannot support the maturation of ADAM17 in all major areas of the mouse brain in the absence of iRhom1 [5]. To learn more about the expression of iRhom2 in the brain, we utilized a reporter mouse, in which the expression of the LacZ gene is driven by the iRhom2 promoter. The iRhom2-LacZ reporter revealed little, if any expression in most of the brain, except for the olfactory bulb, where strong blue staining was evident (Figure 1A). Closer analysis of X-gal-stained sections of the olfactory bulb (OB) showed that the expression of iRhom2 was localized to a subset of glomeruli concentrated in the dorso-medial aspect of the OB (Figure 1B) [25]. The glomeruli of the OB contain the axon terminals of olfactory sensory neurons (OSNs) that originate in the olfactory epithelium (OE). X-gal-stained sections of the OE lining the nasal turbinates revealed high and specific expression of the iRhom2-LacZ reporter in the OSN layer (Figure 1C). The expression patterns of iRhom2 in the OE were independently corroborated by an mRNA in situ hybridization (ISH) analysis on sections of *WT* and *iRhom2-/-* mice (Figure 1D). A similar analysis of iRhom1 expression by mRNA ISH (in situ hybridization) showed little, if any, expression in the OE, except in the submucosal layer of sustentacular cells, in both *WT* and *iRhom2-/-* mice. Moreover, we found similar levels of ADAM17 mRNA in *WT* and *iRhom2-/-* OE (Appendix A). An analysis of publicly available single-cell RNAseq data [26] further confirmed the expression of iRhom2 in mature and immature olfactory sensory neurons (mOSN and iOSNs, Figure 1F,G), whereas little iRhom1 was expressed in these cells (Figure 1H). Other cell types, such as sustentacular cells and horizontal basal cells, also showed the expression of both iRhom1 and iRhom2, consistent with the overlapping expression pattern of both iRhoms in most tissues in mice, except immune cells and the majority of cell types in the brain [5].

To determine when iRhom2 expression first appears in the OE during development, we assessed iRhom2-LacZ expression in the OE of newborn pups (Postnatal day 0, P0) and mice that were 2, 4 or 6 days old. We found minimal expression of the iRhom2-LacZ reporter at P0 but observed gradually increased expression over the first postnatal week from P2 to P6 (Appendix A).

### 2.2. iRhom2-/- Olfactory Epithelium Shows No Histopathologic Defects

Histological analysis of the OE by H&E (hematoxylin and eosin) staining did not show any evident major defects in adult *iRhom2-/-* mice compared to *WT* controls (Figure 2A). There was no significant difference in the pattern or percentage of cells labeled with the mature OSN marker OMP (olfactory marker protein, Figure 2B), immature OSN marker GAP43 (growth assisted protein 43, Figure 2C), the proliferation marker Ki67 (Figure 2D) or the apoptosis marker cleaved caspase-3 (Figure 2E) in *iRhom2-/-* versus *WT* mice. To determine whether iRhom2 affects neurogenesis and/or longevity of OSNs, we injected bromodeoxyuridine (BrdU) intraperitoneally into 8–10 week-old *WT* and *iRhom2-/-* mice. There were no significant differences between the number of BrdU-labeled cells one day after injection and surviving cells at 28 days after injection between *iRhom2-/-* versus *WT* OE (Appendix A).

### 2.3. RNAseq GO Analysis

To investigate the role of iRhom2 in the OE in a high-throughput, unbiased manner, we performed bulk RNAseq on OE tissue isolated from *iRhom2-/-* and *WT* mice at different ages (5, 8, 10 and 30 weeks; 3 mice/ages/genotypes; combined *n* = 12 per genotype). Expression analysis of markers for distinct cell types present in the OE showed no major differences across mature or immature OSN markers (Figure 2F, mOSN, iOSN), consistent with the histological analyses described above.

To determine whether specific biological pathways are altered in *iRhom2-/-* mice, we conducted a gene ontology (GO) analysis. Unexpectedly, the GO analysis showed enrichment of pathways for olfactory receptor activity, G-protein coupled receptor (GPCR) signaling and detection of stimuli involved in smell in the *iRhom2-/-* OE (Figure 3A). As expected, we observed a significant downregulation in genes associated with innate immune responses, likely reflecting the well-characterized consequences of the loss of iRhom2 on macrophage function, including the release of the pro-inflammatory cytokine TNFα [1,2,12] (Figure 3A, Appendix A). The genes contributing to the altered GO pathways are highlighted among other differentially expressed genes (FDR < 0.1, Figure 3B). Cell types that contributed to the downregulated GO terms were mainly non-OSN cells (immune, sustentacular ventral cells, HBCs (horizontal basal cells), Figure 3C, Appendix A). In contrast, contributions to the upregulated GO terms derived mainly from mOSNs and iOSNs (Figure 3D).

### 2.4. Activation of iRhom2/ADAM17 by OR Signaling in a Non-OSN Cell Type

Among the top pathways enriched in *iRhom2-/-* OE, OR activity and GPCR signaling were particularly interesting, since iRhom2 has been implicated in the crosstalk between the GPCR receptor for lipophosphatidic acid (LPA) and EGFR/ERK (extracellular signal-regulated kinases) signaling in mouse embryonic fibroblasts [27]. Moreover, signaling through many GPCRs is known to stimulate the activity of ADAM17 [28,29,30,31,32,33]. In addition, the stimulation of GPCRs in several cell types, including keratinocytes, can activate ADAM17-dependent shedding of EGFR ligands (e.g., HB-EGF, heparin-binding epidermal growth factor like growth factor) and thus the EGFR/ERK pathway [27,29,34] (see model in Figure 4A). ORs are GPCRs, yet it is not known whether ORs activate iRhom2/ADAM17. Primary mouse OSNs are challenging to isolate and culture and we are not aware of available OSN cell lines. Therefore, in order to study whether OR signaling activates iRhom2/ADAM17, we turned to a cell line that expresses an OR with a known odorant ligand and is easy to culture. Specifically, we employed the human keratinocyte HaCaT cell line, which expresses the OR OR2AT4, which can be activated by addition of its odorant ligand, Sandalore. As a positive control for GPCR/ERK crosstalk, LPA (lysophosphatidic acid) was utilized; this is known to activate iRhom2 and ADAM17 [21,25]. Treatment of HaCaTs with LPA (10 µM) for 5 min caused rapid phosphorylation of ERK1/2 that was prevented by the metalloprotease inhibitor marimastat (MM, 5 µM) (Figure 4B). Treatment of HaCaT cells with 1 mM Sandalore for 5 min also triggered rapid phosphorylation of ERK1/2, in agreement with a previous report [35]. ERK phosphorylation induced by Sandalore was blocked by treatment with marimastat (Figure 4B), indicating that OR OR2AT4 activates a metalloprotease, most likely iRhom2/ADAM17, to elicit crosstalk leading to phosphorylation of ERK1/2.

### 2.5. Change in Differential Expression of ORs in iRhom2-/- over Time

Bulk RNAseq revealed that a subset of 14 ORs were significantly upregulated (red dots, FDR (false discovery rate) < 0.05) and 16 ORs were significantly downregulated (blue dots, FDR < 0.05) in the *iRhom2-/-* OE, whereas most ORs (grey dots, >1000 ORs detected) were not differentially expressed (DE) (Figure 5A, *n* = 12 data from combined ages 5, 8, 10 and 30 weeks). We found that the differential expression of these 30 ORs varies across ages (5, 8, 10 and 30 weeks, Figure 5B). A pairwise correlation for ORs that were significantly DE between *iRhom2-/-* and *WT* mice (FDR < 0.05) indicated substantial correlations between 5 and 8 weeks, 8 and 10 weeks, and 10 and 30 weeks, with the strongest correlation observed between the 5- and 8-week samples (Figure 5C). Conversely, minimal correlation was found between 5 and 30 weeks. When we dissected the bulk RNASeq data by mouse age, we noted 23 ORs DE at 5 weeks, 27 ORs DE at 8 weeks, 59 ORs DE at 10 weeks and 39 receptors DE at 30 weeks (Appendix A). The degree of differential expression for ORs changes with age, as can be seen by following the logFC for the DE ORs (FDR < 0.05) at each age over time (Appendix A).

### 2.6. iRhom2 Expression Negatively Correlates with Neuronal Activity

We analyzed a publicly available single-cell RNAseq database of mouse OSNs [36] and stratified OSNs into five subsets, based on the expression level of *iRhom2*, including a subset of OSNs with no detectable *iRhom2* (referred to as *iRhom2-*). In the subset with no detectable *iRhom2* expression, the top upregulated transcripts included the neuronal activity genes *S100a5*, *Dlg2*, *Pcp4l1*, *Kirrel2* and *Lrrc3b* [37,38,39] (Figure 6A). A step-wise inverse correlation exists between OSNs with increasing levels of *iRhom2* and reduced *S100a5* expression (Figure 6B). On UMAP (Uniform Manifold Approximation and Projection), OSNs with the highest *S100a5* expression overlapped with those showing the lowest *iRhom2* expression (Figure 6C). Moreover, cells with no detectable *iRhom2* expression (*iRhom2-*) had higher levels of the activity genes *Dlg2*, *Lrrc3b*, *Pcp4l1* and *Kirrel2* than cells with detectable *iRhom2* (*iRhom2+*, indicating 1 to 5 transcripts per cell, Appendix A). This suggests that odor-induced OSN activation may downregulate *iRhom2*.

To understand why *iRhom2* expression is inversely correlated with OSN activity genes, we examined a dataset generated from single-cell Act-seq of the OSNs of mice that were exposed to the odorant acetophenone or solvent only for 2 h [36]. Single-cell Act-seq is a technique that identifies transcriptomic changes due to neuronal activation. We found *iRhom2* expression significantly decreased (*p* = 0.028), whereas *S100a5* expression significantly increased (*p* < 0.01) in OSNs activated by acetophenone; however, exposure to the control solvent did not significantly affect *iRhom2* expression and had a minimal effect on *S100a5* expression (Figure 6D). Furthermore, analysis of a single-cell RNAseq dataset of OSNs from the open and closed naris of naris occlusion experiments showed that the OSN cluster in UMAP with the lowest *S100a5* activity had the highest *iRhom2* expression and vice versa (Figure 6E). Collectively, these findings suggest that *iRhom2* is downregulated by odor stimulation. However, all five activity genes enriched in OSNs with no detectable *iRhom2* (single-cell RNAseq, see above) were expressed at lower levels in the *iRhom2-/-* OE, with significantly lower expression of *Dlg2*, *Lrrc3b* and *Kirrel2* (Appendix A), suggesting that iRhom2 activity is required for the normal expression of these genes. Taken together, these findings suggest that iRhom2 and most likely ADAM17 have a functional role downstream of OR signaling that may first promote the expression of activity genes but subsequently feeds back to downregulate *iRhom2*.

### 2.7. OSNs Expressing ORs Upregulated in iRhom2-/- Maintain Unusually Stable Gene Expression Amid Environmental Odor Changes

We examined whether OR populations that are upregulated or downregulated in the *iRhom2-/-* OE exhibit unique gene regulatory patterns in response to environmental odor changes using existing scRNAseq data derived from unilateral naris occlusion experiments in mice [36]. To determine the overall transcriptomic changes between the open and closed naris, we conducted pairwise Euclidean distance measurements between every mOSN expressing the same ORs, as outlined in Figure 7A. Euclidean distance is a readout of overall gene expression change in the OSNs between conditions. The Principle Component Analyses (PCA) in Figure 7A provide examples of ORs with relatively small (*Olfrs 1346* and *536*) or large (*Olfrs 1366*, *655* and *1028*) Euclidean distances between the open and closed naris. Most OSNs, including the OR subsets that are downregulated in the *iRhom2-/-* OE (KO- ORs), showed significant gene expression changes upon naris occlusion (Figure 7B). This includes reduced expression of known neuronal activity genes such as *S100a5* (Figure 7C) and increased expression of *iRhom2* (Figure 7D). In contrast, ORs enriched in *iRhom2-/-* (KO+ ORs) have relatively small Euclidean distances (KO+ OR in Figure 7B) and show minimal change in *S100a5* (Figure 7C) and *iRhom2* (Figure 7D) expression despite naris occlusion. In an effort to explore the impact of naris occlusion on the olfactory receptor repertoire, we next examined the fold change in the expression of ORs between nares (Figure 7E). Our findings align with previous research [23], indicating a significant decrease in most OR transcript levels following naris occlusion. However, again, the mOSNs expressing ORs upregulated in the *iRhom2-/-* were the exception to the rule as the group exhibited no significant change in OR transcript levels between the open and closed naris. This phenotype underscores the specialized regulatory mechanisms of the subset of ORs enriched in *iRhom2-/-*, implying their independence from activity-dependent gene expression modifications. Overall, these results point to critical roles of iRhom2 in the fine-tuning of the olfactory landscape and activity in response to odor stimulation and deprivation.

### 2.8. Gene Expression Changes in iRhom2-/- OE Mimic Changes Occuring in Naris Occlusion

Previous work by Tsukahara et al. [36] characterized a subset of genes upregulated by naris occlusion (GEP low) and a subset downregulated by naris occlusion (GEP high). We found that the same subset of genes upregulated by naris occlusion was significantly upregulated in the *iRhom2-/-* OE (Figure 8A). The gene subset downregulated by naris occlusion was also downregulated in the *iRhom2-/-* OE at age 10 weeks and 30 weeks (Appendix A). Moreover, a group of genes described in a review article [40] to be consistently up- or downregulated in an activity-dependent manner across several unilateral naris occlusion studies responded in a similar manner to the inactivation of iRhom2 as in the occluded naris. The genes that are downregulated by activity and upregulated upon naris occlusion are also upregulated in the *iRhom2-/-* OE (Figure 8B). These results indicate that gene expression changes in *iRhom2-/-* OE mirror changes occuring after naris occlusion, further corroborating that iRhom2 regulates activity-dependent gene transcription.

## 3. Discussion

The observation of elevated expression of *iRhom2* in the olfactory bulb led to this study aiming to elucidate the role of iRhom2 in olfactory signaling. The iRhom2-LacZ reporter expression in olfactory bulb glomeruli and OSN cell bodies in the OE confirmed the presence of *iRhom2* in OSNs. Independent verification came from RNA ISH and single-cell RNAseq, showing *iRhom2* mRNA, but not *iRhom1* mRNA, in the OSNs. The selective presence of *iRhom2* in the OSNs was surprising, as *iRhom1* is the primary iRhom in most mouse brain areas [5,18], while *iRhom2* is dominant in immune cells [1,2,12]. *iRhom2* gene knockout caused no evident defects in OE histopathology versus *WT* controls, maintaining similar cell type markers, including immature and mature OSNs and proliferation or apoptosis markers.

Our bulk RNAseq analysis revealed a small subset of 30 ORs that are differentially expressed in *iRhom2-/-* OE (FDR < 0.05), with approximately half upregulated and half downregulated. However, most ORs were not differentially expressed in the *iRhom2-/- OE.* Given that OR level differences in RNAseq are thought to faithfully reflect OSN quantity variances expressing particular ORs in different mouse strains [41], iRhom2 appears to regulate specific OSN abundance in the OE. This change in OR populations might also in turn affect the ability to detect and discern different odors as OR abundance could influence odor threshold [42]. Additionally, the influence of other chemosensory inputs such as trigeminal nerves could also be considered in understanding the role of iRhom2 in chemosensation [43].

The genes upregulated in *iRhom2-/-* OE align with the genes showing increased expression upon naris occlusion, suggesting a connection between odor signaling and iRhom2. The fine-tuning of the OSN repertoire through activity-dependent longevity control and/or OR gene selection is an evolutionary strategy believed to enhance an organism’s environmental adaptability [36]. Stimulating specific ORs can either promote or inhibit their OSN survival [44]. Additionally, odor stimulation can boost the number of newly formed OSNs that express odor-responsive ORs [24]. Enhancing the abundance of particular OSNs is thought to make the OE more responsive to odorant signaling, which can improve the organism’s ability to forage for food, find mates and avoid predators, which are all key activities that offer a survival advantage. In contrast, reducing the abundance of other OSN subsets through odor stimulation is seen as an adaptation mechanism to specific olfactory environments, favoring the survival of non-stimulated neurons for increased sensitivity to new odors and stimuli. These insights, combined with the role of iRhom2 in modulating OR abundance and activity-dependent transcription, indicate multiple signaling pathways activated by odor stimulation and iRhom2 function.

Another perspective involves the development and maturation of the olfactory system. Although the present study used mice with mature olfactory organs (age 5–30 weeks), it is noteworthy that olfactory discrimination capabilities develop prenatally in humans [45] and mice [46]. While fetal and neonatal development is not directly relevant here, it provides context for understanding olfactory system maturation. Investigating the expression pattern and consequences of *iRhom2*-deletion in murine fetuses and neonates could present an interesting avenue for future studies [42].

Interestingly, previously reported single-cell sequencing of OSNs revealed an inverse relationship between *iRhom2* expression and neuronal activity-induced gene upregulation. To reconcile our observation of *iRhom2* expression starting postnatally, presumably upon exposure to air flow or odorants, yet being negatively influenced by neuronal activity in mature OSNs, we propose some possible explanations. Firstly, *iRhom2* expression in embryonic OSNs might be activated by chemicals in the amniotic fluid or postnatally by exposure to air flow or odorants or both. Alternatively, the initial onset of *iRhom2* expression in OSNs during the first postnatal week could be determined by the age of the mouse or of individual OSNs, independent of odor stimulation. Secondly, *iRhom2* expression could become regulated by OR activity as a negative feedback mechanism in adult mice.

How can OR stimulation influence iRhom2 activity and expression? Since ORs are part of the GPCR family, we hypothesize that OR signaling activates iRhom2/ADAM17, similar to the established GPCR-ADAM17-EGFR/ERK signaling pathway [16,27,29,47]. Prior research has highlighted the essential role of metalloproteases, especially ADAM17, in facilitating an interaction between GPCRs and the EGFR, which subsequently regulates cell migration and proliferation [32,34,47,48,49]. The crosstalk between a GPCR, iRhom2/ADAM17 and EGFR forms a triple membrane-spanning signaling pathway, crucial for physiological processes like hair growth, relying on ADAM17-mediated EGFR activation post-GPCR stimulation by LPA [34]. Additionally, several GPCRs have been associated with the ADAM17-driven release of the EGFR ligand TGFα [28]. Despite ORs being the largest GPCR family [20], their involvement in iRhom2/ADAM17-mediated OR signaling within OSNs remains unexplored. Intriguingly, our data and previous studies indicate that OR stimulation by odorants can trigger ERK phosphorylation rapidly [50], suggesting a similar triple membrane-spanning pathway might exist in OSNs. It is also recognized that ORs are expressed in various cells beyond OSNs, such as keratinocytes and macrophages [35,51,52]. This study presents initial evidence that stimulating the keratinocytes OR OR2AT4 enhances ERK activation via a metalloprotease, most likely iRhom2/ADAM17, akin to LPA-induced stimulation [27] (see model in Figure 9). Further studies will be necessary to corroborate whether OR signaling in primary OSNs also activates iRhom2/ADAM17.

Given the known role of iRhom2 in immune cells, it was not surprising that we found that the pathways downregulated in the *iRhom2-/-* OE related to the innate immune response. Since OSNs are not known to have a role in innate immunity, this likely reflects the absence of iRhom2 function in the immune cells of the OE. Since all the mice in this study were kept under homeostatic, non-inflammatory conditions, we would not expect a difference in the OSN health or survival. Future studies will be necessary to determine whether *iRhom2-/-* OSNs may be protected from olfactory inflammation and damage and show enhancement of survival in an olfactory injury model.

Taken together, our studies have identified a role for iRhom2 in modulating the relative abundance of ORs and activity-driven expression profiles, likely through activity-dependent mechanisms. The inverse correlation between *iRhom2* mRNA levels and OSN activity gene expression (e.g., *S100a5*, *Dlg2*, *Kirrel2*) implies that the most active OSNs reduce *iRhom2* expression, possibly to limit their odor responsiveness, as part of an activity-dependent desensitization strategy. Naris occlusion single-cell RNAseq data further reveal that deprivation of OR stimulation in the closed naris causes an increase in *iRhom2* mRNA, potentially sensitizing inactive OSNs. Additionally, specific gene sets identified as upregulated in the occluded naris [36,40] are also elevated in *iRhom2-/-* mice. Together, these findings support a model where OR stimulation activates iRhom2/ADAM17 to process an EGFR ligand (or other proteins), leading to (a) upregulation of OSN activity genes and (b) reduced expression of gene sets that are enriched in the occluded naris (Figure 9). This, in turn, may trigger a negative feedback mechanism that reduces *iRhom2* transcription to desensitize active OSNs. We propose that iRhom2/ADAM17 signaling functions as a rheostat, akin to the model proposed by Tsukuhara et al. [36], where enhanced OR-dependent iRhom2/ADAM17 activity upregulates activity genes but downregulates the expression of iRhom2, maintaining a balance between olfactory sensitization and desensitization.

Why are the subset of ORs that are upregulated in *iRhom2-/-* resistant to the effects of odor deprivation? Intriguingly, OSNs expressing these upregulated ORs in *iRhom2-/-* show fewer gene expression changes upon naris occlusion compared to other OSNs. Thus, the rheostat model might not apply to a small number of atypical OSNs expressing ORs upregulated in *iRhom2-/-*. Rather, our data suggest distinct regulatory models depending on which OR is expressed in a given OSN.

In summary, this study reveals that iRhom2 regulates the OR repertoire of the olfactory epithelium and the gene expression profiles of OSNs in an activity-dependent manner. Odorant stimulation of ORs likely activates iRhom2/ADAM17 catalytic activity and cross-talks in an autocrine fashion to drive intracellular OSN signaling pathways, resulting in the transcriptional regulation of activity-dependent genes. A negative feedback mechanism exists by which active OSNs can downregulate *iRhom2* expression to reduce the sensitivity of the OSNs to activity-dependent changes. iRhom2 may thereby control the sensitization or desensitization of OSNs to odorant stimulation. Additional research will be required to deepen our understanding of iRhom2’s function in modulating OSN abundance and function.

## 4. Materials and Methods

### 4.1. Mice

We employed *iRhom2 KOMP* ([53] Rhbdf2^tm1b(KOMP)Wtsi^, referred to as *iRhom2-/-*, 129Sv, C57BL/6 mixed genetic background) [53], in which the expression of LacZ is driven by the endogenous *iRhom2* promoter.

### 4.2. X-Gal Staining

Adult *iRhom2-/-* or *iRhom1-/-* mice of different ages (as indicated) were sacrificed by CO2 euthanasia for adults or decapitation for neonates following the AVMA Guidelines for the Euthanasia of Animals. Heads were fixed in 4% PFA overnight at 4 °C, decalcified in 0.5 M EDTA, pH 7.4 (Boston Bioproducts, Milford, MA, USA, BM-711) for 2 weeks at 4 °C and cryoprotected in 30% sucrose/PBS for 2 days at 4 °C. Heads were then flash frozen in OCT (Sakura Finetek, Torrance, CA, USA, 4583) and sectioned on a cryostat (Leica Biosystems, Deerpark, IL, USA, CM3050S) using high-profile disposable blades 818 (Leica Biostsystems, Deerpark, IL, USA, 14035838383) at −19 °C at 10 µm thickness. X-gal solution is composed of 100 mmol/L sodium phosphate, 1.3 mmol/L MgCl2, 3 mmol/L potassium ferricyanide, 3 mmol/L potassium ferrocyanide and 1 mg/mL X-gal powder (Apex Bio Tech LLC, Houston, TX, USA, A2539, Batch # 2) dissolved into DMSO, diluted into ddH2O. Sections were stained in X-gal solution for 17 h, washed with ddH2O, counterstained with Eosin B, dehydrated, placed on superfrost plus microscope slides (Cardinal Health, Dublin, OH, USA, M6146-PLUS) and covered with coverslips.

### 4.3. RNA In Situ Hybridization

Specimens from *WT* and *iRhom2-/-* mice 8–12 weeks old were fixed in 10% neutral buffered formalin for 7 days, decalcified in a freshly prepared 14% EDTA pH 7.2–7.4 solution (EDTA 99% pure, Thermo Scientific, Cat # 118432500, Sodium hydroxide 10.0N, LabChem Cat # LC245002) for 21 days, processed routinely in alcohol and xylene, and embedded in paraffin. Five micron thick sections were cut from the paraffin blocks and mounted on glass slides. Chromogenic in situ hybridization was performed on an automated stainer (Leica Bond RX, Leica Biosystems, Deer Park, IL, USA) with RNAscope 2.5 LS Assay Reagent Kit-Red (Advanced Cell Diagnostics, Newark, CA, USA, Cat. # 322150) and Bond Polymer Refine Red Detection (Leica Biosystems, Buffalo Grove, IL, Cat. # DS9390) following the manufacturer’s standard protocol. Slides were stained with appropriate RNA scope 2.5 LS Probes ordered from Advanced Cell Diagnostics (Newark, CA, USA) as follows: Mm-Rhbdf2 (*iRhom2*), Cat No. 476168 (targeting region 760–1667 of NM_172572.3), Mm-Adam17, Cat No. 479518 (targeting region 1383–2325 of NM_009615.6) and Mm-Rhbdf1, Cat No. 476158 (targeting region 414–1421 of NM_010117.2). A positive control probe detecting a housekeeping gene (mouse *Ppib*, Advanced Cell Diagnostics, cat # 313918) and a negative control probe detecting the bacterial (*Bacillus subtilis*) *dapB* gene (Advanced Cell Diagnostics, Cat. # 312038) were used to confirm adequate RNA preservation and detection, and the absence of non-specific signal, respectively. The chromogen was fast red and the counterstain hematoxylin. Positive RNA hybridization was identified as discrete, punctate chromogenic red dots under brightfield microscopy.

### 4.4. Treatment of Keratinocytes with Sandalore and Phospho-ERK Western Blot

HaCaTs were cultured in DMEM with 10%FCS and 1%P/S. Cells were plated at 100,000 cells/well in full media in a 12-well format. The next day, cells were serum starved for 17 h in DMEM without FCS. After starvation, the cells were washed once with pre-warmed PBS and treated with Sandalore (1 mM; Perfumer Supply House, Danbury, CT, USA, CAS#65113-99-7) diluted in DMSO, or DMSO-only vehicle, for 5 min at 37 °C. Cells were removed from the incubator, washed 2X with ice cold PBS and lysed in 100 µL/well of lysis buffer composed of 1% Triton-X100, protease inhibitor cocktail (1:500; Roche, Basel, Switzerland), marimastat (5 uM; Sigma Aldrich, St. Louis, MO, USA, 154039-60-8), 1,10-Phenantroline (10 mM; Sigma Aldrich, St. Louis, MO, USA, P9375-259), Sodium Fluoride (10 mM) and Sodium Orthovanadate (2 mM). Next, 6x SDS sample buffer and 50 mM DTT were added to all samples. Samples were boiled for 7 min at 95 °C. Then, 20 µL of sample was run per lane on 10% SDS-polyacrylamide gels. Gels were transferred in a Trans-blot SD semi-dry transfer cell (BioRad, Hercules, CA, USA, 1703940) onto BioTrace Nitrocellulose membranes (Cytiva, Marlborough, MA, USA, 66485). Membranes were blocked in 5% milk, incubated in p-ERK antibody (1:1000; Cell Signaling Technologies, Danvers, MA, USA, 9101S) or ERK1/2 antibody (1:5000; Sigma Aldrich, St. Louis, MO, USA, M5670) overnight at 4 °C and stained with HRP-conjugated anti-rabbit IgG secondary antibody (1:5000; Promega, Madison, WI, USA, W401B) for 30 min at room temperature. We used an ECL detection system (Cytiva, Marlborough, MA, USA, RPN2106) and a Chemidoc image analyzer (Bio-Rad, Hercules, CA, USA) to expose the blot for 30 s by chemiluminescence.

### 4.5. Immunohistochemistry

IHC was performed on a Leica Bond RX automated stainer using Bond bulk reagents (Leica Biosystems, Buffalo Grove, IL, USA) and a polymer detection reagent kit (DS9800, Novocastra Bond Polymer Refine Detection, Leica Biosystems). The chromogen was 3,3 diaminobenzidine tetrachloride (DAB), and sections were counterstained with hematoxylin. For Ki67, slides were heat induced at pH 9.0 for epitope retrieval, stained with anti-Ki67 (Cell Signaling, Danvers, MA, USA, 12202, 1:500) and secondarily stained with Leica Biosystems DS9800 kit, reagent #3, no dilution. For cleaved caspase-3, slides were heat induced at pH 6.0 for epitope retrieval, stained with anti-cleaved caspase-3 (1:250, Cell Signaling, 9661) and secondarily stained with Leica Biosystems DS9800 kit, reagent #3, no dilution.

### 4.6. Slide Scanning and Cellular/Subcellular Quantification

Slides were scanned by brightfield imaging at 40X magnification on the Axioscan 7 Microscope Slide Scanner (Zeiss, Oberkochen, Germany). Images were analyzed using Qupath 0.4.1 [54]. The olfactory epithelium layer was annotated manually and analyzed. All cells were detected and included in the following analysis using Qupath Cell Detection algorithm. For RNA ISH analysis, transcripts were detected using Qupath Subcellular Detection algorithm with an expected, minimum and maximum spot size of 1.2, 0.5 and 1.2 µm, respectively. The number of dots per cell included individual dots and estimated dot numbers from clusters. For Ki-67 and cleaved caspase-3 quantification, positive cell detection algorithm was used.

### 4.7. BrdU Injection, Tissue Harvest and Staining

BrdU (Sigma-Aldrich, St. Louis, MO, USA, B9285) was diluted to 10 mg/mL solution in sterile PBS and aliquoted for storage at −80 °C. Mice were injected with 100 mg/kg BrdU intraperitoneally and sacrificed either 1 day or 28 days post-injection. Heads were harvested for fixation, decalcification, FFPE and sectioning. Slides were heat induced at pH 9.0 for epitope retrieval, stained with anti-BrdU (Abcam, Cambridge, UK, ab6326, 1:250), post primary stained with Vector Laboratories AI 4001 (1:100) and secondarily stained with Leica Biosystems DS9800 kit, reagent #3, no dilution.

### 4.8. Western Blot Densitometry

The Western Blot images were opened in ImageJ Version 1.53 [55]. Background signal was removed by adjusting the brightness and contrast parameters. Bands were encapsulated by rectangles, labelled with “select first lane” and quantified by analyze > gels > plot lanes. The wand function was used to select all lanes and quantify the densitometry of each band. P-ERK bands were normalized to their corresponding total ERK bands. For representation of data, the untreated, treated and treated + MM conditions were normalized to the treated well (Sandalore or LPA). Statistical comparisons were conducted using Student’s *t*-tests with the *p* value indicated above each of the conditions analyzed.

### 4.9. WOE Dissection, RNA Extraction and Library Prep

Mice were euthanized by CO_2_ and the whole olfactory epithelium (WOE) from both nares was dissected and placed into RNA Later Stabilization Solution (Fisher Scientific, Waltham, MA, USA, AM7020). Tissues were homogenized in Buffer RLT by polytron. The homogenized lysates were transferred to a gDNA Eliminator Spin column, and RNA was extracted using the RNeasy Plus Kit (Qiagen, Germantown, MD, USA, 74134). After the RW1 wash, we performed the DNaseI (Qiagen, Germantown, MD, USA, 79254) on column digestion step for 15 min, followed by a second RW1 wash. All other steps were carried out according to the RNeasy Plus Kit’s standard protocol. Total RNA was submitted to the Weill Cornell Medicine Genomics Core for library prep with the NEB Ultra II Directional RNA Library Prep (plus poly A isolation module). QC was conducted using Agilent Bioanalyzer Sample QC- Nanogel and NanoDrop Spectrophotometer. Samples were sequenced on the NovaSeq6000 (Illumina, San Diego, CA, USA) on an S4 Flow Cell at 2 × 150 cycles.

### 4.10. RNAseq Alignment, Quantification and Differential Expression Analysis

Nextflow nf-core v3.10.1 (https://doi.org/10.5281/zenodo.1400710) was used for reads alignment and quantification. More specifically, sequences were aligned with STAR [56] against GRCm38, and gene-level read quantification was carried out via RSEM [57]. Differential expression analysis between groups was performed against all genes or olfactory receptors only using EdgeR v3.40.2 [58] in R (http://www.R-project.org, accessed on 15 September 2023) and *p*-values were then re-corrected by FDR. Gene nomenclature was retrieved from BioMart [59]. Data analysis, statistical testing and plotting were carried out in python (https://ir.cwi.nl/pub/5008, accessed on 15 September 2023).

### 4.11. Principal Component Analysis

To calculate the principal component analysis (PCA) for our single-cell RNA sequencing data, we utilized the Scanpy library [60]. The AnnData object containing the preprocessed gene expression matrix underwent normalization, scaling and selection of highly variable genes. We then applied the PCA algorithm using the Scanpy function sc.tl.pca, with the svd_solver parameter set to ‘arpack’ to perform the singular value decomposition (SVD). This method projected the high-dimensional data into a lower-dimensional space, retaining the most significant features in terms of variance. The resulting PCA coordinates were stored and used to calculate Euclidean distances.

### 4.12. Visualization via UMAP

The scRNAseq data processing for UMAP dimensional reduction involved initial loading of count data. Quality control steps were implemented to filter cells and genes based on QC metrics, including a minimum of 200 genes per cell and a minimum of 3 cells per gene. Normalization procedures, such as log transformation and identification of highly variable genes, were performed with parameters set to minimum mean count of 0.0125, maximum mean count of 3 and minimum dispersion of 0.5. Unwanted sources of variation, including total counts and mitochondrial gene expression, were regressed out, and the data were scaled and subjected to principal component analysis (PCA) with the number of principal components (PCs) set to 40. Subsequently, nearest neighbor identification and clustering were performed using the Leiden algorithm with a resolution of 0.2, and potential lineage relationships between clusters were inferred via Partition-based Graph Abstraction (PAGA). UMAP was then applied for dimensionality reduction and visualization with the number of PCs set to 40, and clusters were visualized on the UMAP plot colored by metadata variables. Finally, UMAP coordinates were extracted and combined with metadata for visualization of gene expression levels of specific genes of interest on the UMAP plot.

### 4.13. Identification of the OR Expressed in Each OSN

The characterization of OR choice in each mOSN is conducted following the method described previously [36]. Briefly, for each given mOSN, the highest expressing OR is determined to be the sole expression OR.

### 4.14. OR Pair-Wise Comparison

The pairwise comparison of OR was conducted across two different measurements: individual gene expression and Euclidean distance comparison between open and closed nares. Firstly, the pairs of OR comparison are obtained by generating all matches of open and closed naris cells expressing the same OR. First, for each given pair, the expression levels of OR genes were analyzed between open and closed nares using log-fold change calculations, providing insights into differential expression patterns. Secondly, Euclidean distance measurements were employed to assess the overall similarity or dissimilarity in gene expression profiles between cells from open and closed nares. This analysis enabled the characterization of the spatial relationships between cells in the scRNAseq dataset. Violin plots were generated to visualize the distributions of log-fold changes, facilitating the comparison of gene expression patterns across OR subsets that were upregulated and downregulated in the *iRhom2-/-*.

### 4.15. OR Population Comparison

We examined each OR gene individually, assessing the counts of cells associated with open and closed nares to gauge relative abundance of OR with and without odor stimulation. The log1p was calculated between nares to capture the fold change in counts between total number of cells from closed and open nares. This transformation offers a nuanced representation of fold changes without addressing issues with extreme values. To ensure comparability across receptors, we normalized the log1p values based on the proportion of counts each OR contributes to the total counts across all cells. This normalization procedure standardized the effect size (fold change) to account for variations in counts across ORs, facilitating the identification of fold change with significant count alterations between open and closed nares. Ultimately, these normalized log1p fold change values were used in characterizing the relative counts of OR genes in response to naris occlusion.

### 4.16. RNAseq Data Analysis

RNAseq analyses were performed in python (versions 3.8) and R (version 4.2). Single-cell analyses were performed in python (versions 3.8) using Scanpy package (version 1.8.2) [60]. Custom scripts for data analysis and visualization were built using open-source python libraries (pandas, numpy, matplotlib, plotly, sklearn, scipy, seaborn and itertools). Scripts to replicate data analysis are available at https://github.com/Justice-Lu/iRhom2_Analysis (accessed on 15 September 2023).

## Figures and Tables

**Figure 1 ijms-25-06079-f001:**
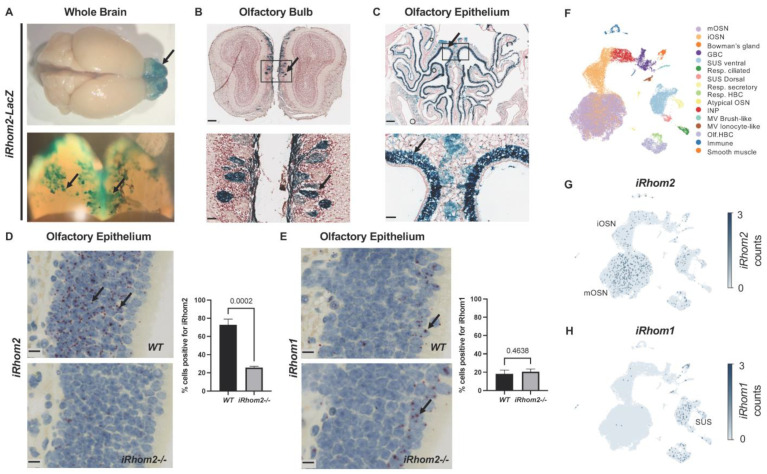
Characterization of the expression of iRhom2 and iRhom1 in olfactory sensory neurons (OSNs). (**A**) Whole brain from iRhom2-LacZ mice stained by X-gal reaction revealed that most brain tissue is negative for iRhom2-expression except for a region in the olfactory bulb (OB, black arrow). (**B**) Sections of the OB of iRhom2-LacZ stained by X-gal reaction revealed iRhom2 reporter expression in some, but not all, OSN glomeruli (indicated by black arrow), with expression mainly in the dorsal-medial section of the OB (scalebar = 250 µm upper panel, 50 µm lower panel). (**C**) Sections of the olfactory epithelium (OE) of iRhom2-LacZ stained by X-gal reaction revealed iRhom2 reporter expression in OSN cell bodies (black arrow, scalebar = 250 µm upper panel, 50 µm lower panel). (**D**) mRNA in situ hybridization (ISH) of olfactory epithelium with probe Mm-Rhbdf2 confirmed high iRhom2 mRNA levels in the OSNs of WT (pointed by black arrows), but not iRhom2-/- mice (scalebars = 10 µm). iRhom2-/- mice have no functional iRhom2 protein, but express exons 1 and 2 as part of the targeting constructs [20], resulting in low levels of transcript presence in the iRhom2-/- OE. (**E**) mRNA ISH staining of OE with probe Mm-Rhbdf1 showed that no iRhom1 is expressed in WT OSNs, although iRhom1 is expressed in the sustentacular cells (indicated by black arrows, scalebars = 10 µm). (**F**–**H**) Analysis of single-cell RNAseq data of OE cells showed high expression of iRhom2 in immature and mature OSNs (iOSN, mOSN) (**F**,**G**), but minimal expression of iRhom1 in these cells (**H**). iRhom1 is expressed in non-OSN cells in the OE, such as sustentacular cells (SUS).

**Figure 2 ijms-25-06079-f002:**
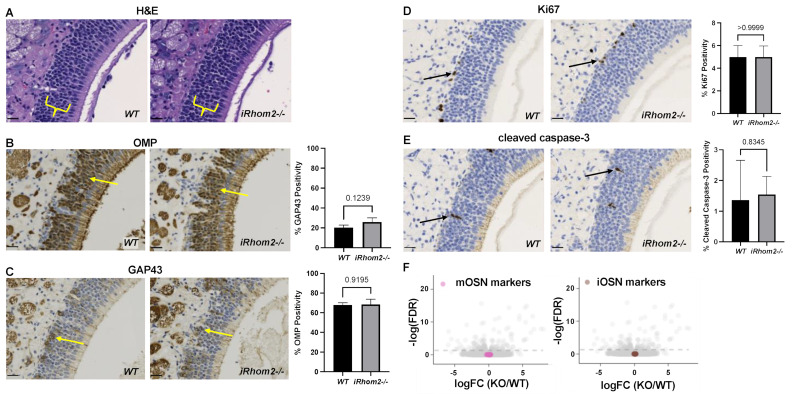
Histologic and immunohistochemical comparison between *WT* and *iRhom2-/-* olfactory epithelium revealed no apparent phenotypic differences. (**A**) Sections stained with H&E revealed no evident difference in the appearance of the OSN cell body layer (yellow brackets) or axon bundles (scalebars = 20 µm). (**B**,**C**) Staining with OMP to mark mature OSNs (**B**, yellow arrows) or GAP43 to mark immature OSNs (**C**, yellow arrows) and subsequent quantification revealed no differences in the number of mature or immature OSNs, respectively (scalebars = 20 µm). (**D**,**E**) Staining of OE sections with antibodies against the proliferation marker Ki67 (**D**, black arrows) or against cleaved caspase-3 to identify apoptotic cells (**E**, black arrows) revealed no significant differences in proliferation or apoptosis, respectively (scalebars = 20 µm). (**F**) RNAseq analysis of OE isolated from *WT* and *iRhom2-/-* mice (*n* = 12 each) showed no difference in the expression of markers for mature OSN (mOSN) or immature iOSN (iOSN).

**Figure 3 ijms-25-06079-f003:**
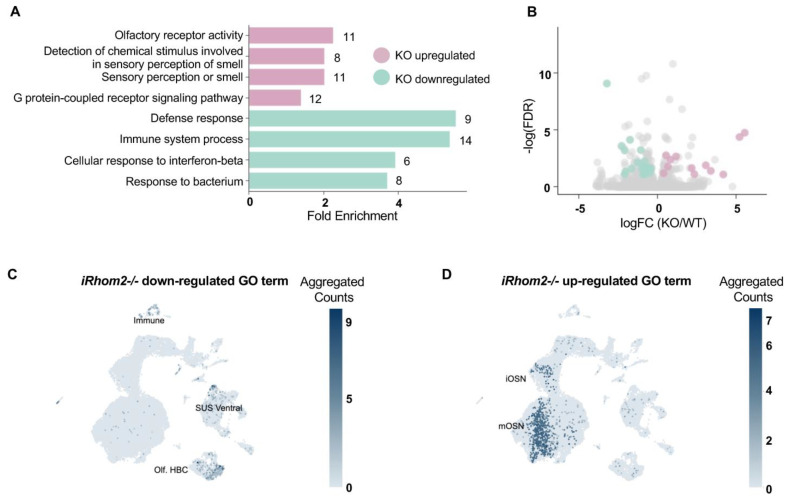
Bulk RNAseq GO analysis of *iRhom2-/-* OE versus *WT* OE. (**A**) GO analysis revealed that pathways upregulated in the *iRhom2-/-* OE were exclusively related to OR/GPCR signaling, while pathways downregulated were related to innate immune function. (**B**) Volcano plot of genes differentially expressed between WT and *iRhom2-/-*. Individual genes that contributed to the GO terms are labelled in pink (upregulated) and green (downregulated) plotted by the logFC and −log(FDR). Only genes with FDR < 0.1 were considered for the GO analysis, all others are labelled in gray. (**C**,**D**) Aggregated counts of the top 4 GO terms in WT and *iRhom2-/-* were plotted on a UMAP of OE cell types. Downregulated GO terms in the *iRhom2-/-* OE are predominantly localized to non-neuronal tissues, including immune cells (**C**). Upregulated GO terms in the *iRhom2-/-* OE are predominantly localized to mOSNs and iOSNs (**D**).

**Figure 4 ijms-25-06079-f004:**
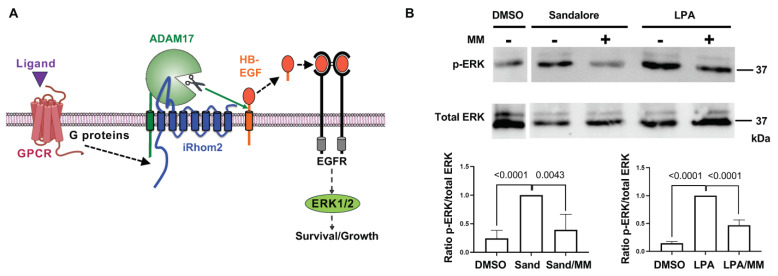
Crosstalk between the OR OR2AT4 on keratinocytes and ERK1/2. (**A**) Schematic depicting the triple-membrane spanning pathway, by which GPCR signaling activates iRhom2/ADAM17, which in turn results in the shedding of HB-EGF and activation of EGFR/ERK1/2 and stimulation of cell survival, growth or proliferation. (**B**) Stimulation of human HaCaT cells with Sandalore (Sand) stimulates phosphorylation of ERK1/2 (p-ERK), which can be prevented by addition of the metalloprotease inhibitor marimastat (MM, 5 µM), which blocks the activity of ADAM17 (**B**). Stimulation of HaCaT cells with the GPCR-agonist LPA results in a similar increase in pERK1/2, which can be blocked by MM (5 µM). Densitometric quantification (lower panels) show significant reduction in p-ERK signal of the DMSO or MM conditions compared to Sandalore- or LPA-treated conditions (results of Student’s *t*-test indicated, *n* = 4).

**Figure 5 ijms-25-06079-f005:**
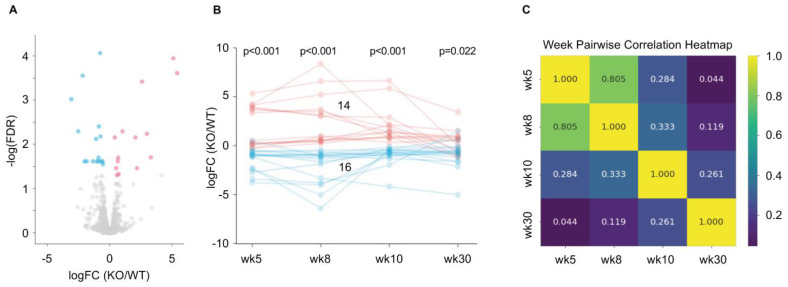
Differentially expressed (DE) ORs in *iRhom2-/-* OE versus *WT* OE. (**A**) Volcano plot shows the distribution of DE ORs upregulated (14 ORs, red) or downregulated (16 ORs, blue) in the *iRhom2-/-* OE with FDR < 0.05 (*n* = 12 per genotype). Most ORs are not differentially expressed in the *iRhom2-/-* OE (>1000 ORs, grey). (**B**) Line graph showing the expression of upregulated (red) or downregulated (blue) ORs across ages at week 5, 8, 10 and 30. Results of Student’s *t*-tests are indicated above each time point. (**C**) A pairwise Pearson correlation of OR logFC across the different time points shows higher correlation (r) of DE ORs at closer age ranges and lower correlation between larger age ranges.

**Figure 6 ijms-25-06079-f006:**
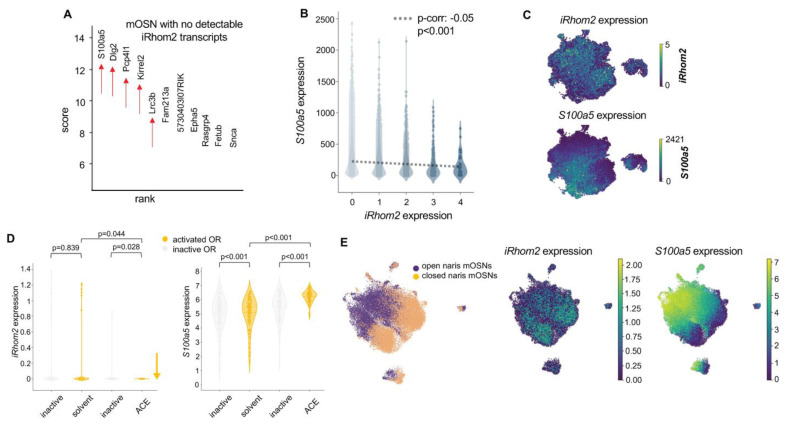
*iRhom2* expression inversely correlates with activity genes. (**A**) Based on single-cell RNAseq data analysis of mOSN cells with no detectable *iRhom2* expression, the 5 top enriched transcripts are known OSN activity genes (*S100a5*, *Dlg2*, *Pcp4l1*, *Kirrel2, Lrc3b*, pointed to by red arrows in (**A**)). (**B**) There is an inverse correlation between the levels of *iRhom2* and *S100a5* expression. (**C**) UMAP shows that OSNs in the region with highest *S100a5* expression correlate to the OSNs with the lowest *iRhom2* expression. (**D**) Analysis of single-cell RNAseq data from Tsukahara et al. [36] shows that *iRhom2* expression is significantly reduced upon acetophenone (ACE) exposure for 2 h, whereas *S100a5* expression is significantly increased upon ACE exposure. The solvent is the vehicle control dipropylene glycol. (**E**) UMAP from single-cell RNAseq of OSNs dissected from naris occlusion experiments. Data are integrated computationally while retaining their open or closed naris tags to create a single UMAP (left panel). *iRhom2* expression is most prominent in the OSNs from the closed naris (middle panel), while *S100a5* expression is most prominent in the OSNs from the open naris (right panel).

**Figure 7 ijms-25-06079-f007:**
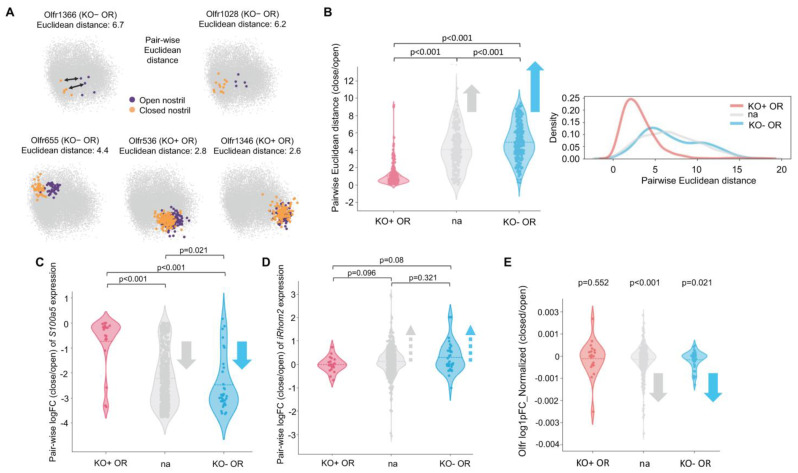
Odor-dependent regulation of *iRhom2* and *S100a5* expression. (**A**) Principal Component Analysis (PCA) of single-cell RNAseq data from the naris occlusion experiment by Tsukahara et al. [36]. Scheme of pairwise Euclidean distance comparison (top left, indicated by black double-headed arrows) and representative PCA for ORs enriched in *iRhom2-/-* OE (KO+ OR) and ORs depleted in *iRhom2-/-* OE (KO- OR). The pairwise Euclidean distance quantifies the overall transcriptional difference between mOSNs in the closed and open naris that express the same ORs. *Olfrs 1346* and *536* are examples of KO+ ORs with short Euclidean distances (2.6 and 2.8, respectively), representing minimal transcriptional differences upon naris occlusion. *Olfrs 1366*, *655* and *1028* are examples of KO- ORs with relatively large Euclidean distances (6.7, 4.4 and 6.2, respectively), representing larger transcriptional differences upon naris occlusion. (**B**) Violin plots and kernel density estimate (kde) plots of Euclidean distances show that KO+ ORs (red) cluster around low Euclidean differences, whereas KO- ORs (blue) have higher Euclidean differences. ORs that are not DE in *iRhom2-/-* are labeled as ‘na’ (gray). (**C**) KO+ ORs show minimal *S100a5* expression change in naris occlusion versus all other ORs, which have large negative fold changes in *S100a5*. (**D**) KO+ ORs show minimal changes in *iRhom2* expression upon naris occlusion versus all other ORs, which upregulate *iRhom2*. (**E**) KO+ ORs show minimal changes in OR expression between nares, while all other ORs show negative fold change in OR expression with naris occlusion. Gray and blue arrows indicate the direction of change.

**Figure 8 ijms-25-06079-f008:**
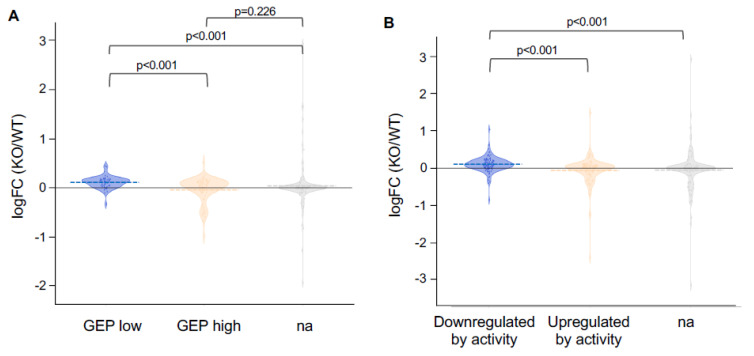
Gene sets upregulated by naris occlusion are also upregulated in *iRhom2-/-* OE. (**A**) The gene set characterized by Tsukahara et al. [36] as being upregulated upon naris occlusion (GEP low) is also upregulated in the *iRhom2-/-* OE versus *WT* OE of *n* = 12 per genotype and significantly different (*p* = 0.0) from the gene set downregulated on naris occlusion (GEP high) or not significantly affected (na). (**B**) Similar observations were made with gene sets downregulated by odor activity as characterized by the meta-analysis by Wang et al. [40], which were upregulated in the *iRhom2-/-* OE versus *WT*.

**Figure 9 ijms-25-06079-f009:**
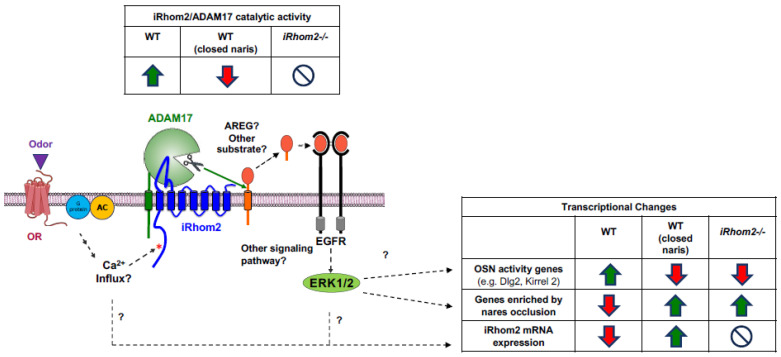
Schematic of OR/iRhom2/ADAM17 pathway resulting in catalytic activity of the iRhom2/ADAM17 protease complex and transcriptional changes in activity genes, naris occlusion genes and iRhom2. Question marks indicate currently unknown mechanisms, the red asterisk next to the cytoplasmic domain of iRhom2 indicates possible phosphorylation upon OR activation, and the circle with a backslash symbol indicates that *iRhom2* mRNA or iRhom2 protein are not present in the *iRhom2-/-* mice.

## Data Availability

The raw and processed RNA-Seq datasets generated as part of this study are available from NCBI GEO at GSE268171. The single-cell RNAseq of mouse olfactory epithelium is publicly available at GSE151346. An additional single-cell RNAseq dataset of mOSNs of wildtype and unilateral naris occlusion is also publicly available at GSE173947.

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
