# Peer review of "Role of iRhom2 in Olfaction: Implications for Odorant Receptor Regulation and Activity-Dependent Adaptation"

_ijms, 2024, doi:10.3390/ijms25116079_

Round 1
Reviewer 1 Report
Comments and Suggestions for Authors
The manuscript by Azzopardi et al. entitled “Role of iRhom2 in Olfaction: Implications for Odorant Receptor Regulation and Activity-dependent Adaptation” reports that iRhom2 is uniquely expressed in olfactory sensory neurons (OSNs) and suggests a mechanism by which odor stimulation of OSNs activates iRhom2/ADAM17 catalytic activity, resulting in downstream transcriptional changes to the OR repertoire and activity genes, and driving a negative feedback loop to downregulate iRhom2 expression
The topic it is very interesting. In fact, understanding the mechanisms underlying the neurogenesis of OSNs and the expression of ORs, and the factors that can modulate it is a topic still to be fully investigated.
However, there are some aspects that require further attention from the authors:
1) Why did the authors choose this experimental model? Why did they choose the sandal OR2AT4? I find that the purpose is not well explained and therefore remains incomprehensible to the reader.
2) Relationship with TNFalpha levels. It seems to me that in the model used the levels of TNFalpha are reduced, so the survival of the OSN should also be greater. How do the authors explain this?
3) What do the authors mean by beneficial odors? What are beneficial smells? This aspect should be explained better.
Reviewer 2 Report
Comments and Suggestions for Authors
This manuscript is a murine study of primary olfactory receptors in the nasal mucosa with focus upon iRhom2, a binding partner for the cell surface metalloprotease ADAM17. The first part of the Abstract and of the Introduction define well what this binding receptor molecule is and that it is uniquely expressed in primary olfactory sensory neurons. The Abstract might also specify the various techniques used in this study.
In the Discussion, in addition to defining the role in sensitivity and olfactory threshold, it might be useful to mention olfactory discrimination of different odours. Even if the authors do not yet have sufficient evidence for a definitive statement in this regard, its importance could briefly be mentioned, In addition, it would be useful to comment on whether other types of sensory input might alter the role of iRhom2. For example, ammonia has a very distinctive repugnant odour that is easily identified and not confused with any other (in humans), but ammonia also is an irritant that simultaneously stimulates trigeminal nerve pain endings in the olfactory mucosa. Garlic is another example, with an additional function in taste. Could painful stimuli thus change the threshold function of olfactory receptors?
Another perspective not addressed in this study but that could be mentioned briefly in the Discussion is the aspect of development and maturation of the olfactory system, even though all mice in the present study were young adults with mature brains (age range 8-12 weeks as stated under the heading “RNA in-situ hybridization”, though these data and the number of animals examined would be better state in the first paragraph of “Materials and Methods”). The development of the olfactory bulb including synaptogenesis in the human has been described, as well as available data on functional olfactory discrimination even prenatally as early as 30 weeks gestation (Sarnat HB, Yu W. Maturation and dysgenesis of the human olfactory bulb. Brain Pathol 2016;26:301-318). Whereas fetal and neonatal development of the olfactory system is not germaine to the present study, it provides a certain perspective that enables better understanding. Indeed, if data are available about the timing of expression of iRhom2 in the murine fetus and neonate, these data would be appropriate to mention in the Discussion.
The manuscript is well written with good English grammar. References are appropriate in number and selection. Figures are of good quality. In Figure 1, the stain or immunoreactivity should be mentioned in the legend for each of the subfigures. The supplementary file is satisfactory.
